# Identification of microRNAs Responding to Aluminium, Cadmium and Salt Stresses in Barley Roots

**DOI:** 10.3390/plants10122754

**Published:** 2021-12-14

**Authors:** Liuhui Kuang, Jiahua Yu, Qiufang Shen, Liangbo Fu, Liyuan Wu

**Affiliations:** 1Department of Architectural Engineering, Yuanpei College, Shaoxing University, Shaoxing 312000, China; kuangliuhui@zju.edu.cn; 2Key Laboratory of Crop Germplasm Resource of Zhejiang Province, Department of Agronomy, Zhejiang University, Hangzhou 310058, China; robert_jiahua@163.com (J.Y.); shenqf@zju.edu.cn (Q.S.); fulb@zju.edu.cn (L.F.)

**Keywords:** abiotic stresses, microRNA, barley, target genes, aluminum, cadmium, salinity

## Abstract

Plants are frequently exposed to various abiotic stresses, including aluminum, cadmium and salinity stress. Barley (*Hordeum vulgare*) displays wide genetic diversity in its tolerance to various abiotic stresses. In this study, small RNA and degradome libraries from the roots of a barley cultivar, Golden Promise, treated with aluminum, cadmium and salt or controls were constructed to understand the molecular mechanisms of microRNAs in regulating tolerance to these stresses. A total of 525 microRNAs including 198 known and 327 novel members were identified through high-throughput sequencing. Among these, 31 microRNAs in 17 families were responsive to these stresses, and Gene Ontology (GO) analysis revealed that their targeting genes were mostly highlighted as transcription factors. Furthermore, five (miR166a, miR166a-3p, miR167b-5p, miR172b-3p and miR390), four (MIR159a, miR160a, miR172b-5p and miR393) and three (miR156a, miR156d and miR171a-3p) microRNAs were specifically responsive to aluminum, cadmium and salt stress, respectively. Six miRNAs, i.e., miR156b, miR166a-5p, miR169a, miR171a-5p, miR394 and miR396e, were involved in the responses to the three stresses, with different expression patterns. A model of microRNAs responding to aluminum, cadmium and salt stresses was proposed, which may be helpful in comprehensively understanding the mechanisms of microRNAs in regulating stress tolerance in barley.

## 1. Introduction

Abiotic stresses such as salinity, heavy metals and aluminum (Al) in soil severely limit crop production worldwide, with estimated losses of over USD 120 billion p.a. [1]. At present, nearly 50% of the world’s potential arable lands are acidic, and Al toxicity is a primary factor affecting crop growth in acid soil [2]. Al (mainly in the form of Al^3+^ in acidic soil) is toxic to plant growth by targeting various cellular sites and processes, which leads to the inhibition of root elongation and the uptake of water and nutrients [2,3,4]. Heavy metal contamination in soil, mainly due to unwise industry activities and the application of metal-containing fertilizers, has raised a severe threat to crop production and food safety [5]. Cadmium (Cd) is commonly considered one of the most widespread and dangerous heavy metals. Cd stress not only inhibits plant growth and development, leading to reduced yield, but also causes the potential damage to human health through the food chain [6]. Moreover, soil salinization is also a major abiotic factor affecting crop planting and production worldwide [7]. Currently, over 900 million hectares of soil have been affected by salinity worldwide, which is becoming a bottleneck of sustainable development for agriculture [7].

Barley (*Hordeum vulgare*) is an important cereal crop, ranking fourth in terms of planting area in the world, and it is characterized by wider adaptation and higher salt tolerance when compared to other cereal crops [8,9]. Actually, barley also shows wide genetic variation in its tolerance to Al and Cd toxicity [10,11]. Moreover, barley is an excellent model crop in physiological and molecular studies, especially for the mechanisms of tolerance to various stresses in plants.

Over the past few years, more and more studies have been focused on efficient and precise gene regulation mediated by microRNAs (miRNAs). miRNAs are endogenous non-coding RNA molecules with approximately 20–24 nucleotides (nt) in length, which bind with target mRNA sequences through complementary base pairing, leading to the cleavage of target genes [12]. Currently, 7057 miRNAs have been identified in 73 plant species (http://www.mirbase.org, accessed on 08 January 2019). It is well known that miRNAs play diverse roles in many intracellular activities, including development, metabolism and stress defense [12]. For instance, osa-miR393 is associated with primary and crown root growth in rice [13] and miR172-guided *HvAPETALA2* (*HvAP2*) mRNA cleavage determines the cleistogamous flowering and grain density in barley [14,15]. Furthermore, over-expression of miR393 increases the sensitivity to salt and drought stresses due to the reduced expression of the target genes *OsTIR1* (*transport inhibitor response 1*) and *OsAFB2* (*auxin-signaling F-box 2*) in rice [16]. Transgenic rice and *Arabidopsis thaliana* plants constitutively over-expressing osaMIR396c have shown reduced salt and alkali stress tolerance [17]. Similarly, miRNAs are also involved in the responses to heavy metals such as chromium (Cr) and Cd [18,19]. Over-expression of miR166 in rice could decrease Cd translocation from roots to shoots and accumulation in grains, and enhance Cd tolerance through regulating the target gene *OsHB4* (homeodomain-containing protein) [19]. Moreover, miRNAs involved in the response to Cd, Al or salt stress have been identified in cultivated and wild barley [20,21,22]. However, all of the previous studies were focused on the functioning of miRNAs in regulating gene expression under a specific abiotic stress, and little research has been carried out on miRNA-mediated gene networks in plants under multiple abiotic stresses, which occurs frequently for field crops.

In this study, a barley cultivar, Golden Promise, which is widely used in genetic transformation, was treated with or without Al, Cd or salt stress. Small RNA libraries were developed to identify miRNAs responding to single or multiple stresses and degradome sequencing was performed to discover their target genes. We hypothesized that these differently expressed miRNAs and their targets played significant roles in tolerance to Al, Cd and salt stresses in barley. These identified specific miRNAs could be useful for breeding new barley cultivars with high tolerance to abiotic stresses.

## 2. Results

### 2.1. Identification of Conserved and Novel miRNAs in Golden Promise

The root growth of the barley cultivar Golden Promise was dramatically inhibited under Al, Cd and salt stress treatments in comparison with controls (Figure 1a–c). In order to identify miRNAs responding to the three stresses, 12 small RNA libraries were prepared and sequenced. The clean reads were BLASTN-searched in miRBase to identify the conserved and novel miRNAs. As a result, a total of 525 miRNAs were detected, including 198 known miRNAs, 94 new members of known miRNAs and 233 potential candidate miRNAs. Of these, miRNAs of 21 nt in length were most dominant (Figure 2a). The majority of miRNAs with lengths of 18–22 nt started with a 5′ U whereas the first nucleotide was A base for most 24–25 nt miRNAs (Figure 2b).

According to the BLASTN search results, 246 miRNAs were categorized into 47 known miRNA families (Figure 3). There was a large variation in the members of each miRNA family, with 11 conserved families, such as MIR156, MIR166 and MIR167_1, containing more than 10 miRNA members. Moreover, there were 16 families which consisted of only one member, such as MIR394, MIR528, MIR8005, and so on.

### 2.2. Identification of miRNAs Responding to Al, Cd and Salt Stresses

The miRNAs with fold-changes of larger than 0.5 were considered to be differentially expressed miRNAs under stress conditions. A total of 31 differentially expressed miRNAs were found in barley roots in response to one, two or all of three stresses (Table 1 and Appendix A). Among these, five miRNAs were upregulated and 12 were downregulated in response to Al stress. Under Cd stress, 13 miRNAs were upregulated and seven were downregulated. Under salt stress, seven and 11 miRNAs were up- and downregulated, respectively. Furthermore, 12 miRNAs specifically responded to only one stress. For example, ata-miR156a-3p and ata-miR156d-3p were specifically responsive to salt stress. Under Al stress, hvu-miR166a and ata-miR166a-3p were specifically depressed. The miRNA hvu-MIR159a-5p was upregulated and ata-miR160a-5p was downregulated under Cd stress. In total, there were 12 miRNAs responding commonly to two stresses. For instance, ata-miR169c-5p, ata-miR169c-5p and osa-miR319a-3p.2-3p were remarkably upregulated under both Cd and salt stresses. In addition, there were six miRNAs which responded commonly to all of three stresses, including ata-miR156b-3p, ata-miR166a-5p, ata-miR169a-3p, ata-miR171a-5p, ata-miR394-5p and ata-miR396e-5p. However, most of these multiple stresses-responsive miRNAs differed in the expression pattern under the different stresses. For example, miR171a and miR394 were significantly upregulated when exposed to Cd toxicity, but were obviously downregulated under Al and salt stress. Only the miRNA ata-miR156b-3p showed the same expression pattern in responding to the three stresses (Table 1 and Appendix A).

### 2.3. Characterization of Target Genes for miRNAs

Putative targets for 290 miRNAs were detected via degradome analysis and bioinformatics prediction (Appendix A). As a result, 136 miRNAs were found to be associated with 213 target genes based on bioinformatics predictions, and other 154 miRNAs were related to 287 targets based on degradome analysis (Appendix A). Interestingly, many target genes were transcription factors, suggesting their essential roles in regulatory networks under abiotic stress responses. For example, *TCP family transcription factor 4* (*TCP4*) was negatively targeted by osa-miR319a-3p.2-3p, which was upregulated under Cd and salt stresses. In addition, ata-miR167a-5p was downregulated under Al and Cd stresses, and could regulate *Auxin response factor 6* (*ARF6*). Furthermore, a number of target genes encoded receptor kinase and calmodulin, such as hvu-MIR159a-5p-mediated *Mitogen-activated Protein Kinase 16* (*MPK16*) and ata-miR1432-5p-regulated *Calmodulin like 43* (*CML43*), as well as EF hand calcium-binding protein (Appendix A).

### 2.4. GO Analysis of Target Genes Regulated by Differently Expressed miRNAs

A total of 70 putative target transcripts were found for 31 differently expressed miRNAs under three stresses. Among these, 56 target genes were associated with GO analysis in three ontologies. In terms of biological processes, two enriched categories—transcription (DNA-templated) and regulation of transcription (DNA-templated)—occupied the largest components under all three stresses (Figure 4). As for molecular function, transcription factor activity (sequence-specific DNA binding) and DNA binding were most dominant under various stresses. For cellular components, most of the target genes were localized in the nucleus. Obviously, the current results indicate that miRNAs play important roles in the regulatory network of Al, Cd and salt stress responses, mainly via the modulation of the target transcription factors.

## 3. Discussion

The physiological and molecular mechanisms of tolerance to various abiotic stresses have been intensively investigated in barley [23,24,25]. However, there are few reports focusing on the miRNA-mediated networks involved in abiotic stress tolerance. In the present study, 525 miRNAs were identified in barley roots exposed to three stresses (Figure 2a). Among these, 246 miRNAs could be classified into 47 known miRNA families (Figure 3). The analysis of miRNA expression profiles showed that 31 miRNAs were dramatically responsive to one, two or three stresses (Table 1 and Appendix A). Furthermore, GO analysis showed that the majority of targets for these miRNAs were transcription factors (Figure 4). Based on the obtained results, we proposed a putative model of miRNA-involved regulatory networks for Al, Cd and salt stress responses in barley roots (Figure 5).

### 3.1. miRNAs Specifically Responding to Al Stress

In this study, hvu-miR166a and ata-miR166a-3p were downregulated only under Al stress, which targeted the transcripts of homeobox leucine zipper family genes (Table 1 and Appendix A). HD-Zip IIIs (a group of homeobox transcription factors) were involved in many developmental processes, such as lateral root growth and shoot meristem formation in *Arabidopsis* [26,27]. Moreover, miR166-mediated cleavage of *HD-Zip III* was involved in the tolerance to abiotic stresses such as drought in rice [28]. The expression of downstream genes contributing to cell wall formation was altered in the miR166 knockdown lines of rice [28]. The root cell wall (CW) was recognized as the major target cell component of Al toxicity and accumulated more than 90% of absorbed Al [4]. Therefore, it may be assumed that hvu-miR166a and ata-miR166a-3p play an important role in copying with Al stress through regulating the genes relevant to cell wall formation and modification in barley.

In addition, the expression of ata-miR390-5p was also specifically repressed under Al stress in barley roots, similarly with that observed in *Medicago truncatula* [29]. miR390 targeted the genes encoding Leucine-rich repeat receptor-like protein kinases (LRR-RLKs), which acted as receptors of hormones such as abscisic acid (ABA), which are crucial in the transduction of plant environmental signals [30,31]. As one of the topmost transcription factors in ABA signaling transduction pathways, ABI5 positively participated in the Al stress response in *Arabidopsis* and rice beans (*Vigna umbellate*) through regulating a series of downstream genes associated with cell wall structure and modification [32]. It was also found that ABA content was increased in barley roots exposed to Al stress [32,33,34]. Therefore, it may be suggested that miR390 is involved in Al stress tolerance by mediating ABA signal transduction pathways associated with cell wall modification (Figure 5).

### 3.2. miRNAs Specifically Responding to Cd Stress

The root elongation of *Arabidopsis* was notably reduced under Cd stress, but lateral root formation was significantly promoted [35]. Meanwhile, Cd stress reduced the endogenous auxin level and the induced toxicity could be alleviated by supplying exogenous auxin in *Arabidopsis* by stimulating the synthesis of hemicellulose 1 and increasing the cadmium fixation capacity of root cell walls [35,36]. In this study, ata-miR393-5p was up-regulated after Cd exposure, which negatively regulated two auxin receptor genes, *HvTIR1* and *HvAFB*, resulting in changes of downstream *auxin response factor*
*(**ARF**)* family members in the auxin signaling pathway [16]. ata-miR160a-5p and ata-miR172b-5p were both downregulated under Cd stress in barley roots, and their targets were referred to *ARF* genes through degradome sequencing analysis (Table 1 and Appendix A). Recently, it was shown that ARF7 was firstly activated by auxin signaling, and then modulated the MPK3/MPK6 cascade, hence promoting the expression of cell wall remodeling genes associated with pectin degradation and cell wall separation, consequently leading to lateral root emergence [37]. In barley, hvu-MIR159a-5p displayed a higher expression level under Cd stress, targeting the gene *HORVU1Hr1G088510.1* encoding Mitogen-activated protein kinase 16 (Table 1 and Appendix A). The signaling modules of MPK cascades are highly conserved in eukaryotes [38]. Thus, it may be assumed that the auxin signaling pathway, combined with MPK modules, is involved in the lateral root emergence of barley in response to Cd stress [35,36,37].

### 3.3. miRNAs Specifically Responding to Salt Stress

Salt stress positively regulated the *GA 2-oxidase* family gene *GA2ox7*, leading to the reduction of bioactive gibberellin (GA) [39]. Moreover, *OsGA2ox5* over-expressing rice lines showed higher resistance to salinity than wild-type plants [40]. In this study, ata-miR156a-3p was obviously upregulated under salt stress and its target gene *HORVU3Hr1G072810.1* encoded Gibberellin 2-oxidase. Moreover, degradome sequencing analysis showed that another miR156 family member, ata-miR156d-3p, was upregulated under salt stress, targeting the gene encoding UDP-glycosyltransferase (UGT) superfamily protein (Table 1 and Appendix A). Two glycosyltransferase genes, *AtUGT79B2* and *AtUGT79B3,* were proven to be positive regulators of drought, low-temperature and as salt stresses by promoting anthocyanin accumulation [41]. Anthocyanin could serve as an antioxidant, scavenging ROS in response to stresses [42]. Obviously, these two miR156 members are involved in salt stress responses, regulating gibberellin and anthocyanin accumulation in barley, respectively (Figure 5).

### 3.4. miRNAs Responding to Various Abiotic Stresses

A number of miRNAs responding to two or three stresses were also identified. The miRNA ata-miR167a-5p was downregulated under either Al or Cd exposure, and its target gene was *ARF6* based on degradome analysis. The *Arabidopsis* double mutant of *arf10*/*16* showed significant enhancement in Al tolerance due to modification of the cell wall [43]. Thus, the miR167a/ARF6 module is probably associated with Al tolerance by modifying the barley cell wall, and Cd stress by activating the aforementioned auxin signaling pathway combined with MPK signaling modules (Figure 5).

Moreover, ata-miR1432-5p, targeting the genes encoding calmodulin-like 43 and EF hand calcium-binding protein, responded to both Al and salt stresses in the different expression patterns. Calcium is recognized as a second messenger and its sensor proteins contain calmodulin (CaM) and calcineurin B-like protein (CBL) [44]. The *cbl1 Arabidopsis* mutant was more sensitive to Al stress, partially due to differently expressed genes associated with cell wall modification [45]. Furthermore, reduced GA content was also observed in *cbl1* mutants [46]. Hence, it may be assumed that the miR1432/calcium-binding protein module alleviates Al toxicity by modifying the cell wall and enhances salt tolerance through reducing GA accumulation (Figure 5).

Lateral root development could be promoted by miRNA164-directed cleavage of *NAC1* in maize [47]. In the current study, ata-miR164a-5p was downregulated under either Cd or salt stress. Its target gene, *HORVU2Hr1G080460.8*, encoded the NAC domain-containing protein, implying the similar regulation of lateral root growth in barley. Unlike ata-miR164a-5p, osa-miR319a-3p.2-3p was upregulated under these two stresses. It was reported that over-expression of miR319 could reduce the level of jasmonic acid in tomato [48]. Moreover, jasmonic acid treatment enhanced salt tolerance in *Brassica napus* [49]. *Arabidopsis* mutants with functional deficiency in terms of jasmonic acid synthesis showed more sensitivity to Cd than the wild-type plant due to elevated expression of the genes associated with Cd uptake [50]. On the other hand, miR319-mediated *TCP4* could positively regulate miR396, which was upregulated under Al and Cd exposure but downregulated under salt stress. miR396 over-expressing transgenic plants exhibited the reduced expression of its targets, *growth-regulating factors* (*GRFs*), resulting in suppressed root growth [51]. Therefore, it can be hypothesized that miR319 and miR396 participate in Al, Cd and salt tolerance through regulating root growth, whereas miR319 is also involved in Cd and salt tolerance by modulating jasmonic acid synthesis in barley (Figure 5).

## 4. Materials and Methods

### 4.1. Plant Culture and Stress Treatment

To observe the root growth under different stresses, the seeds of barley cultivar Golden Promise were disinfected with 3% H_2_O_2_ for 20 min, washed with floating tap water and soaked in deionized water for 2 h. Then seeds were transferred to wet filter papers in germination boxes in a dark environment (22/18 °C, day/night). On the third day, light was supplied. Parts of the seedlings were transferred into 1 mM CaCl_2_ solution (pH 4.5) containing 10 μM AlCl_3_ and the solution was renewed daily. The remaining seedlings were grown in germination boxes for 7 days, and then cultured in the aerated one-fifth Hoagland solution (pH 6.0, renewed every 3 days) for 5 days. Finally, twelve-day-old seedlings were treated with one-fifth Hoagland solution containing 200 mM NaCl and 5 μM CdCl_2_, respectively. The solution without AlCl_3_, NaCl or CdCl_2_ additions was used for controls. After stress exposure for 10 days, the length of the longest root was measured.

To generate miRNA data, seeds were disinfected and germinated following the aforementioned method. After 7 days’ germination, seedlings were transferred into 5-L plastic containers with aerated one-fifth Hoagland solution. Salt treatment was initiated at 7 days after transplanting by adding NaCl at a rate of 100 mM per day, to reach a final concentration of 200 mM in the solution. After 10 days of salt treatment, roots from salt-treated and control conditions were harvested. Twelve-day-old seedlings were treated with one-fifth Hoagland solution containing 5 μM CdCl_2_ and root samples were collected after 10 days. Meanwhile, twelve-day-old seedlings were exposed to 1 mM CaCl_2_ solution (pH 4.5) containing 10 μM AlCl_3_. The CaCl_2_ solution (pH 4.5) without Al was considered as the control. The roots under control and Al-treated conditions were harvested after 24 h and frozen immediately in liquid nitrogen, then stored at −80 °C. All of the root samples, including three treatments and corresponding controls, were finally used for small RNA and degradome sequencing analysis.

### 4.2. Construction of Small RNA and Degradome Libraries

In addition to these 12 small RNA libraries (root tissue of Golden Promise in controls and three stresses with two biological replicates), three degradome libraries (three stresses and their corresponding controls were mixed well, respectively) were also constructed to obtain the targets of miRNAs. Trizol reagent (Invitrogen, Carlsbad, CA, USA) was used to extract the total RNA of samples and then methods described previously were performed for the construction of two kinds of libraries [22,52]. To construct small RNA libraries, TruSeq Small RNA Sample Prep Kit (Illumina, San Diego, CA, USA) was used. Adenylated single-stranded 3′ and 5′ adapters were ligated to small RNA respectively with the use of T4 RNA ligase 2. The reverse transcription reaction was performed to synthesize the first-strand cDNA with subsequent PCR amplification. Finally, PCR products with the length of 140–160 bp were collected from 6% polyacrylamide Tris-borate-EDTA gel. To prepare the degradome library, the methods described by Ma et al. [52] were performed with some modifications. The poly(A) RNA was used as the input RNA and annealed with biotinylated random primers. Strapavidin capture of RNA fragments and 5′ adaptor ligation to RNAs containing 5′-monophosphates were carried out. Then reverse transcription and PCR amplification were performed. Two kinds of libraries were single-end sequenced (50 bp) on an Illumina Hiseq2500 platform (Illumina, San Diego, CA, USA).

### 4.3. Data Processing of Sequencing

The raw data of high-throughput small RNA sequencing for the root samples from Al, Cd and salt stresses and their controls were processed using the ACGT101-miR program (LC Sciences, Houston, TX, USA) to remove adapters and sequences without typical miRNA features. Then, sequences were mapped to mRNA (http://plants.ensembl.org/Hordeum_vulgare/Info/Index, accessed on 1 January 2021), Repbase (http://www.girinst.org/repbase, accessed on 1 January 2021) and RFam (http://rfam.janelia.org, accessed on 1 January 2021) databases to eliminate mRNA, prototypic sequences representing repetitive DNA (repeats) and common non-coding RNA families such as rRNA and snRNA, excepting miRNAs. Subsequently, the remaining clean sequences with 18–25 nucleotides were matched to miRBase (version 21) through a BLAST search. The miRNAs that were perfectly or imperfectly (within two mismatches) mapped to mature miRNAs of plant species were recognized as known miRNAs. The miRNAs which did not match to miRNAs but successfully mapped to the opposite arm of known precursors were defined as new members of known miRNAs. Potential candidate miRNAs could not be matched to known mature miRNAs or precursors, and generally showed low expression. Furthermore, the genomic location of each identified miRNA was determined when the matched pre-miRNAs were BLASTed against the barley reference genome.

Bioinformatics analysis was conducted using TargetFinder to identify miRNA binding sites. Moreover, degradome libraries were constructed to obtain the targets of miRNAs. Pipeline (version 1.5; Illumina) was used to filter raw sequencing reads and then sequencing data were analyzed by the software package CleaveLand3.0 [53]. The sequences of degradome libraries were BLASTed against the barley mRNA database in IPK. The links of all databases listed in this study were shown in our previous research [21]. In addition, GO annotation analysis was carried out using Blast2GO software (https://www.blast2go.com/, accessed on 1 February 2021) based on mRNA sequences.

### 4.4. Characterization of miRNAs Responding to Al, Cd and Salt Stresses

The miRNAs in response to Al, Cd and salt stresses were expected to meet the criteria described by Wu et al. [21]. Moreover, miRNAs were detected in all libraries and at least one normalized readcount was greater than 100 in these three stress treatments or controls. The fold change of normalized reads between stress treatment and control samples was calculated using the formula: fold change = log_2_ (stress reads/control reads). The miRNAs were defined as upregulated if fold change ≥ 0.5, and downregulated if fold change ≤ −0.5.

## 5. Conclusions

In the current study, 31 miRNAs and their targets were identified in the roots of barley seedlings exposed to Al, Cd and salt stresses. Among these, 12 miRNAs were specifically responsive to only one stress, such as miR390 responding to Al stress, miR393 responding to Cd stress and miR156d responding to salt stress, whereas there were several miRNAs, such as miR169, miR396 and miR1432, which responded to multiple stresses. These miRNAs and their target genes participated in the regulation of root growth and development via various signal transduction pathways under abiotic stresses. A model of miRNAs responding to Al, Cd and salt stresses was proposed, which may provide an overall understanding of the mechanisms of miRNAs in regulating stress tolerance in barley.

## Figures and Tables

**Figure 1 plants-10-02754-f001:**
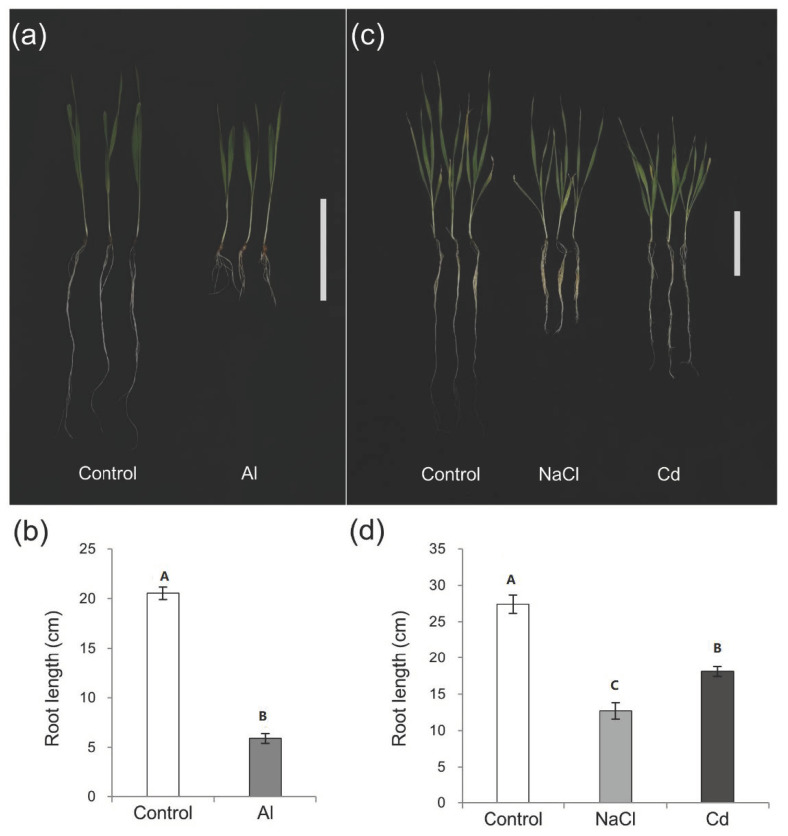
The root growth of the barley cultivar Golden Promise under different conditions. Three-day-old seedlings were exposed to 10 μM AlCl_3_ for 10 days (**a**,**b**). Twelve-day-old seedlings were treated with 5 μM CdCl_2_ and 200 mM NaCl (**c**,**d**) or control conditions for 10 days, respectively. Bar = 10 cm. Data are means ± SD (n = 6), and significant difference (*p* < 0.05), compared via Tukey’s HSD test.

**Figure 2 plants-10-02754-f002:**
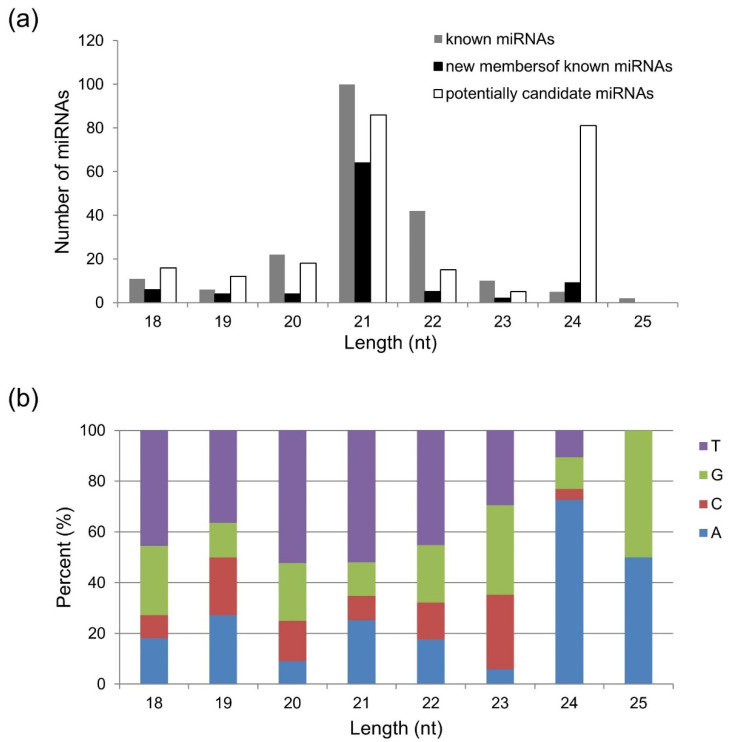
Overview of miRNAs identified in barley roots. (**a**) Length distribution of miRNAs in three classes. (**b**) The prevalence of first nucleotide bias in miRNAs with different lengths.

**Figure 3 plants-10-02754-f003:**
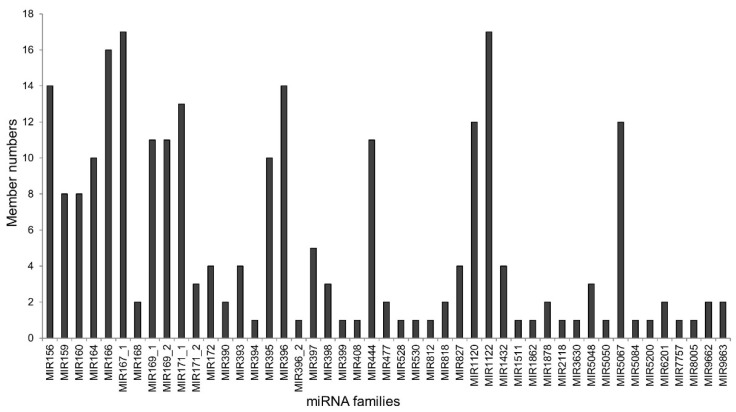
Member numbers of miRNA families.

**Figure 4 plants-10-02754-f004:**
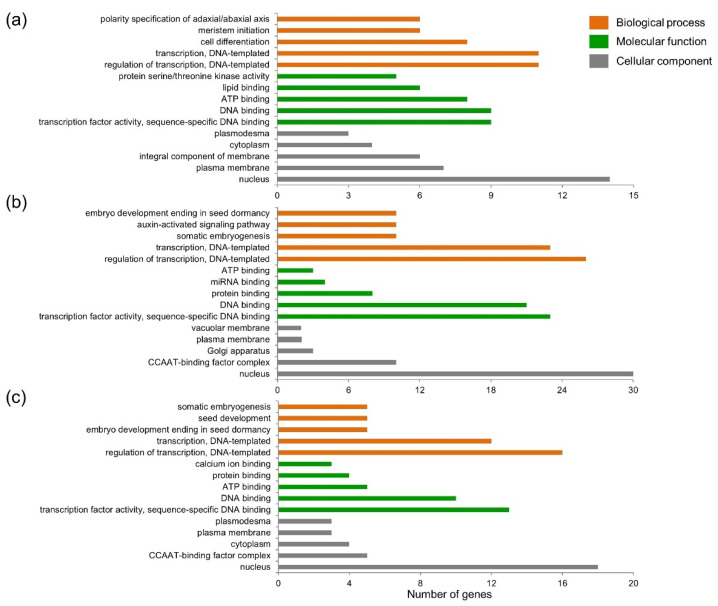
Gene ontology analysis of the putative target genes for 31 differently expressed miRNAs. The categorization of putative targets for miRNAs responsive to Al (**a**), Cd (**b**) and salt (**c**) stress was performed according to biological processes, molecular functions and cellular components.

**Figure 5 plants-10-02754-f005:**
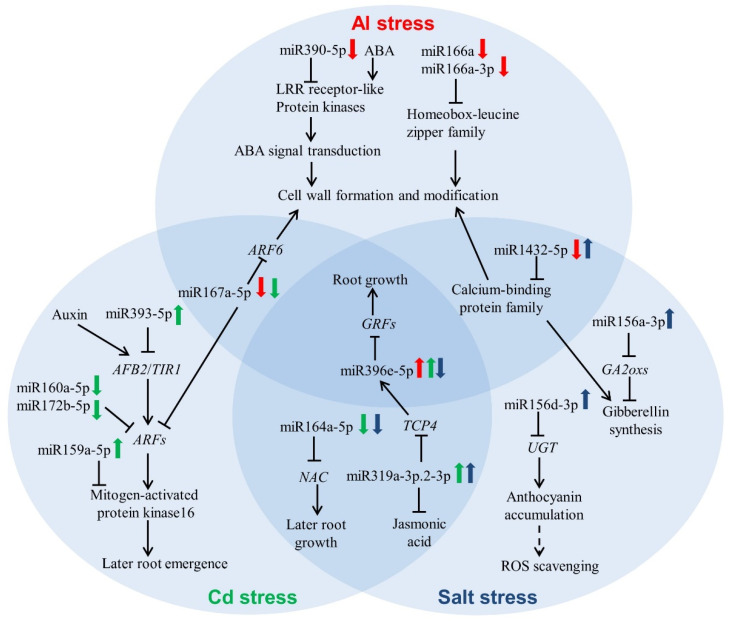
A proposed model of miRNAs involved in Al, Cd and salt responses in barley roots. Downregulation of miR390-5p, miR166a and miR166a-3p promotes the process of cell wall formation and modification, thus alleviating Al detoxification. Upregulation of miR393-5p and miR159-5p can repress later root emergence through controlling the module of the auxin signaling pathway combined with MPK under Cd conditions. To alleviate root inhibition, miR160a-5p and miR172b-5p are significantly downregulated, thus increasing the expression of *ARFs*. Two miR156 family members, miR156a-3p and miR156d-3p, are downregulated, probably leading to reduced ROS scavenging and enhanced gibberellin synthesis after salt treatment. Contrary to Al and Cd treatments, repressed miR396e-5p under salt conditions promotes root growth through the accumulation of *GRFs*.

**Table 1 plants-10-02754-t001:** Differentially expressed miRNAs in response to Al, Cd and salt stresses in barley roots.

miRNA Name	Al ^a^	Cd ^b^	Salt ^c^	Target Rranscript	Annotation	Degradome Detection ^d^
ata-miR156a-3p	0.20	0.05	0.85	HORVU3Hr1G072810.1	Gibberellin 2-oxidase	+
				HORVU1Hr1G078160.1	P-loop containing nucleoside triphosphate hydrolases superfamily	+
ata-miR156b-3p	1.15	0.71	1.95	HORVU6Hr1G028980.8	Cinnamoyl coa reductase 1	-
ata-miR156d-3p	−0.10	0.35	1.26	HORVU5Hr1G042740.2	UDP-Glycosyltransferase superfamily	+
hvu-MIR159a-5p	−0.44	0.56	0.38	HORVU1Hr1G088510.1	Mitogen-activated protein kinase 16	-
osa-miR319a-3p.2-3p	−0.35	1.79	1.09	HORVU2Hr1G060120.1	TCP4	+
				HORVU5Hr1G103400.1	TCP4	+
ata-miR160a-5p	−0.44	−0.65	−0.20	HORVU2Hr1G089670.2	ARF10	+
				HORVU2Hr1G089660.7	ARF10	+
				HORVU7Hr1G101270.6	ARF16	+
				HORVU6Hr1G026750.1	ARF18	+
				HORVU1Hr1G041770.6	ARF22	+
ata-miR164a-5p	−0.39	−1.32	−1.13	HORVU2Hr1G080460.8	NAC domain protein	+
				HORVU7Hr1G072670.3	NAC domain containing protein 1	+
				HORVU5Hr1G011650.2	NAC domain containing protein 1	+
				HORVU5Hr1G041400.1	Phytosulfokine 2	+
hvu-miR166a	−0.82	0.30	0.01	HORVU5Hr1G010650.1	Homeobox-leucine zipper family	+
				HORVU5Hr1G061410.29	Homeobox-leucine zipper HOX10	+
				HORVU0Hr1G010250.3	Homeobox-leucine zipper HOX32	+
				HORVU1Hr1G041790.2	Homeobox-leucine zipper family	+
ata-miR166a-3p	−0.54	0.06	0.22	HORVU1Hr1G041790.2	Homeobox-leucine zipper family	+
				HORVU5Hr1G010650.1	Homeobox-leucine zipper family	+
ata-miR166a-5p	−0.71	1.06	−0.99	HORVU1Hr1G076940.1	Nucleotide-diphospho-sugar transferase family	-
				HORVU6Hr1G005350.2	GPI mannosyltransferase 3	-
ata-miR167a-5p	−0.63	−0.84	−0.10	HORVU2Hr1G121110.32	ARF6	+
ata-miR167b-3p	−0.92	1.17	0.13	HORVU1Hr1G075520.2	Jacalin-related lectin 3	-
ata-miR167b-5p	0.58	−0.26	−0.33	HORVU2Hr1G059280.1	SWI/SNF complex subunit SWI3C	-
				HORVU2Hr1G059130.1	SWI/SNF complex subunit SWI3C	-
tae-miR167c-5p	0.64	1.20	0.20	HORVU1Hr1G077630.2	Ubiquitin carboxyl-terminal hydrolase 25	-
				HORVU2Hr1G059280.1	SWI/SNF complex subunit SWI3C	-
				HORVU2Hr1G059130.1	SWI/SNF complex subunit SWI3C	-
ata-miR167f-3p	0.29	1.69	1.79	HORVU4Hr1G016990.3	Cysteine desulfurase	-
hvu-miR168-3p	−0.71	0.19	−0.68	HORVU5Hr1G037570.4	Receptor-like protein kinase	-
				HORVU4Hr1G031620.1	14-3-3 protein beta/alpha-A	-
hvu-miR168-5p	−0.55	−0.16	−1.02	HORVU1Hr1G055570.4	WD repeat-containing protein WRAP73	+
				HORVU2Hr1G105050.1	Protein of unknown function (DUF581)	+
ata-miR169a-3p	0.58	−2.90	−6.21	HORVU4Hr1G087430.2	Unknown	-
ata-miR169c-5p	−0.37	−2.27	−∞	HORVU5Hr1G092700.17	NF-YA10	+
				HORVU4Hr1G075830.4	NF-YA3	+
				HORVU6Hr1G081080.12	NF-YA5	+
				HORVU2Hr1G032130.27	NF-YA5	+
				HORVU2Hr1G032130.6	NF-YA5	+
ata-miR169i-5p	0.42	−2.04	−2.89	HORVU5Hr1G092700.17	NF-YA10	+
				HORVU4Hr1G075830.4	NF-YA3	+
				HORVU6Hr1G081080.12	NF-YA5	+
				HORVU2Hr1G032130.27	NF-YA5	+
				HORVU2Hr1G032130.6	NF-YA5	+
hvu-miR171-3p	0.28	0.72	−1.31	HORVU6Hr1G063650.1	GRAS	+
				HORVU1Hr1G053510.1	GRAS	+
ata-miR171a-3p	−0.36	0.09	−1.40	HORVU4Hr1G061310.1	GRAS	+
ata-miR171a-5p	−1.42	0.62	−1.08	HORVU2Hr1G076620.7	T-complex protein 11	+
ata-miR172b-3p	−0.63	−0.10	−0.17	HORVU5Hr1G112440.1	Ethylene-responsive TF10	+
				HORVU1Hr1G011800.24	AP2-like ethylene-responsive TF	+
ata-miR172b-5p	−0.19	−1.45	−0.03	HORVU7Hr1G106280.1	ARF6	+
				HORVU6Hr1G088570.2	Clathrin interactor EPSIN 2	+
ata-miR390-5p	−1.86	0.10	0.23	HORVU7Hr1G007520.1	LRR-RLK	-
				HORVU1Hr1G043790.1	LRR-RLK	-
				HORVU2Hr1G091840.16	RLK2	-
				HORVU2Hr1G124010.6	RLK	-
ata-miR393-5p	−0.16	1.02	0.03	HORVU2Hr1G070800.3	HvAFB	+
				HORVU1Hr1G021550.4	HvTIR1	+
ata-miR394-5p	−1.18	0.83	−0.63	HORVU1Hr1G043940.3	Protein TIC110, chloroplastic	+
				HORVU6Hr1G018370.1	Calnexin 1	+
ata-miR396e-5p	0.51	0.62	−1.36	HORVU7Hr1G008680.14	GRF5	+
				HORVU4Hr1G010080.6	GRF6	+
				HORVU4Hr1G003440.12	GRF9	+
ata-miR1432-5p	−1.56	0.00	0.84	HORVU1Hr1G094160.1	Calmodulin like 43	+
				HORVU5Hr1G111520.1	EF hand calcium-binding protein family	+
tae-MIR9662a-5p	−0.38	1.16	0.63	HORVU5Hr1G123930.2	Beta-fructofuranosidase, insoluble isoenzyme 3	-
				HORVU2Hr1G100080.7	Protein strawberry notch homolog 1	-

^a^ Al, ^b^ Cd and ^c^ Salt represent the fold-changes between different treatments (Al, Cd and salt stresses) and their control-normalized reads. This was calculated using the formula: fold change = log2 (stress reads/control reads). −∞ indicates that miRNA expression was not detected in Golden Promise under salt treatment. ^d^ Degradome detection indicates that the target genes of miRNAs were either detected (+) or not (-) in the degradome sequencing results. The “MIR” in the miRNA name means that the miRNA belongs to new member of known miRNAs.

## Data Availability

The raw data are available at NCBI database with the BioProject accession numbers PRJNA481620, PRJNA485436 and PRJNA507337.

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
