# Peer review of "Identification of microRNAs Responding to Aluminium, Cadmium and Salt Stresses in Barley Roots"

_plants, 2021, doi:10.3390/plants10122754_

Round 1

Reviewer 1 Report

According to the authors, the findings of their study, when combined with the degradome sequencing analysis, reveal that a putative regulatory network of miRNAs is connected with the tolerance of barley roots to Al, Cd, and salt stresses. Overall, manuscript is well written I have some concerns.
Abstract must be rewritten with more detail information.
Please start each sentence with full form, no matter you are using abbreviations.
Even though the authors provide an extensive literature
review of the Introduction, they should enhance and reconstruct it in the introduction section. They should provide the scope of the paper clearly.
In general, there is a repetition of information that might have been omitted.
Check the English language of the results and discussion section.
Discuss every section in detail. Most of the sections are left open.
Update conclusion section in a meaningful way.

Reviewer 2 Report

The aim of this study is to develop a small RNA library in order to identify miRNAs responding to a single or multiple stresses. The paper is well written and deserves to be published. There are some minor suggestions:

The introduction could be focussed more on the identification of microRNAs.

Reviewer 3 Report

Review of the manuscript ID: plants-1481964

Title: “ Identification of microRNAs responding to aluminum, cadmium and salt stresses in barley roots”

The manuscript describes the identification of 31 new miRNA for Cd, Al, and salt stress in roots of barley (Hordeum vulgare) grown in hydroponic conditions. Specific miRNA (5,5,3) were found for Cd, Al, and salt stress respectively affecting the plant cell wall structure, the jasmonate and ABA signalling pathways. The majority of the other 31 new miRNA were for transcription factors.

The work is interesting for the study of plant stress although the conditions of the experiment are far from a realistic scenario.

There is one obscure point in the methods section :

Lines 256-260: it appears that the Cd and the salt stress were administered at the same time, while in the whole manuscript they were treated as separately administered. Please clarify this important point.

Line 270: For the analysis of the degradosome the authors followed Ma et al 2010, did the Author use the software package CleaveLand to accomplish all the calculations related to the data elaboration?

In the section on Results, why didn’t the authors produce a heatmap of the up and down-regulated miRNA in the three conditions?

In Table 1, what does the number below the miRNA sequence mean? Please insert their meaning in the table caption.

In the Discussion,

Line 145-146: this sentence needs a reference.

Line 150: The Authors should be careful when they indicate that some miRNAs are present under double or triple stress. As a fact, the Authors did not perform experiments with actually double or triple stresses, or so they affirm, therefore the miRNA found out are only present in the combination of two and three stress separately, which is a different fact altogether.

Lines 193-194: Define what ARF stands for when talking about genes

Lines 202-203: this assertion need a reference for backup

Lines 2011-2015: the presence of secondary metabolites such as anthocyanin does not mean that there is oxidative stress going on and that ROS are present. Moreover, the authors did not measure the level of oxidative stress in the plant roots due to the treatment, nor the amount of possible anthocyanin produced into the roots (or even better into the whole plant). Also in Figure 5 the presence of ROS is all hypothetical.

The Conclusion content does not vary from that of the abstract.

There are two blearing spell mistakes, one in the title where Aluminium is misspelled and another in the abstract where at line 10 there is “Barely” instead of barley. In the abstract, the full scientific name of barley, Hordeum vulgare, should have been mentioned.

There are four references that are far too old:

Ref 26 and 27 both from 1993 and 5 and 6 from 1995. I am sure there are more recent references available for the same purposes. Moreover, Ref 8, Su et al., 2014, does not display the Figures in the online version, here again, please find a substitute.

Round 2

Reviewer 1 Report

Authors have significantly improved the manuscript, therefore it can be accepted for publication.

Reviewer 3 Report

I would like to thank the Authors for addressing punctually all my requests and concerns and for having modified the text accordingly.

The manuscript has been greatly improved and I consider it ready for publication.